# easyXpress: An R package to analyze and visualize high-throughput *C. elegans* microscopy data generated using CellProfiler

**Joy Nyaanga**[1,2], **Timothy A. Crombie**[1], **Samuel J. Widmayer**[1], **Erik C. Andersen**[1]*

1 Department of Molecular Biosciences, Northwestern University, Evanston, IL, United States of America,
2 Interdisciplinary Biological Sciences Program, Northwestern University, Evanston, IL, United States of America

* erik.andersen@gmail.com

**Data Availability Statement:** All data are contained within the manuscript and are also available in a public repository: https://github.com/AndersenLab/easyXpress.

## Abstract

High-throughput imaging techniques have become widespread in many fields of biology. These powerful platforms generate large quantities of data that can be difficult to process and visualize efficiently using existing tools. We developed easyXpress to process and review *C. elegans* high-throughput microscopy data in the R environment. The package provides a logical workflow for the reading, analysis, and visualization of data generated using CellProfiler's WormToolbox. We equipped easyXpress with powerful functions to customize the filtering of noise in data, specifically by identifying and removing objects that deviate from expected animal measurements. This flexibility in data filtering allows users to optimize their analysis pipeline to match their needs. In addition, easyXpress includes tools for generating detailed visualizations, allowing the user to interactively compare summary statistics across wells and plates with ease. Researchers studying *C. elegans* benefit from this streamlined and extensible package as it is complementary to CellProfiler and leverages the R environment to rapidly process and analyze large high-throughput imaging datasets.

## Introduction

Developments in high-throughput imaging techniques have led to a rapid increase in these data. Researchers are able to move away from the laborious manual collection of images that typically limits large-scale analyses [1]. Furthermore, these advances have enabled scientists to collect data of intact cells, tissues, and whole-organisms with increased temporal and spatial resolution [2]. However, typical users require software methods for efficient handling, analysis, and visualization to make the most of these extensive image datasets.

*C. elegans* is a globally distributed, free-living roundworm nematode that is amenable to many types of experimental biology. The *C. elegans* cell lineage is completely characterized [3], and the *C. elegans* connectome is completely mapped [4], making these animals an exemplary model for developmental biology and neurobiology. The species can also be rapidly reared in large, genetically diverse populations in laboratory settings, providing unparalleled statistical power for experimental biology compared to any other metazoan [5]. Furthermore, metabolic

**Funding:** J.N. and E.C.A. received support from the NSF-Simons Center for Quantitative Biology at Northwestern University (awards Simons Foundation/SFARI 597491-RWC and the National Science Foundation 1764421). This project and equipment was funded by an NIH grant (ES029930) from the National Institute of Environmental Health Sciences to E.C.A.

**Competing interests:** The authors have declared that no competing interests exist.

and developmental pathways in *C. elegans* are conserved in humans [6]. High-throughput imaging technologies can improve *C. elegans* studies by increasing experimental efficiency, scalability, and quality. Existing systems for automated image acquisition, such as the Molecular Devices ImageXpress platforms generate images of nematodes that can be analyzed with software like CellProfiler's WormToolbox [7] to extract nematode phenotype information. This software uses probabilistic nematode models trained on user selected animals to automate the segmentation of nematodes from the background of images in high-throughput. As a result, CellProfiler's WormToolbox is able to measure hundreds of phenotypes related to animal shape, intensity, and texture. Implementing this software for large-scale imaging experiments can generate large quantities of data that requires additional analysis software for reliable and reproducible handling, processing, and visualization. CellProfiler Analyst was developed to offer tools for the analysis of image-based datasets, but this software is not integrated with modern statistical environments. We sought to design a resource that facilitates the exploration of CellProfiler data in the R environment [8], where this limitation can be eliminated. The R language provides extensive open-source statistical and data visualization tools that are well supported by the user community. In leveraging R, we are able to create a flexible tool that can be rapidly integrated with other statistical R packages to suit project-specific analysis needs.

We developed easyXpress, a software package for the R statistical programming language, to assist in the processing, analysis, and visualization of *C. elegans* data generated using CellProfiler. easyXpress provides tools for quality control, summarization, and visualization of image-based *C. elegans* phenotype data. Built to be complementary to CellProfiler, this package provides a streamlined workflow for the rapid quantitative analysis of high-throughput imaging datasets.

## Methods

### Preparation of animals for imaging

Bleach-synchronized animals were fed *E. coli* HB101 bacteria suspended and allowed to develop at 20˚C with continuous shaking. Animals in 96-well microtiter plates were titered to approximately 30 animals per well. Prior to imaging, animals were treated with sodium azide (50 mM in 1X M9) for 10 minutes to paralyze and straighten their bodies.

### Imaging

Animals in microtiter plates were imaged at 2X magnification with an ImageXpress Nano (Molecular Devices, San Jose, CA). The ImageXpress Nano acquires brightfield images with a 4.7 megaPixel CMOS camera and are stored in 16-bit TIFF format. The images were processed using CellProfiler software (for details see https://github.com/AndersenLab/CellProfiler).

### Paraquat dose response

A 1.5 M solution of paraquat (Methyl viologen dichloride, Sigma, 856177-1G) was prepared in sterile water, aliquoted, and frozen at -20˚C until used. Experimental animals were grown at 20˚C and fed OP50 bacteria spotted on modified nematode growth medium, containing 1% agar and 0.7% agarose to prevent animals from burrowing. After three generations of passaging, animals were bleach-synchronized and embryos were transferred to the wells of 96-well microplates. Each well contained approximately 30 embryos in 50 μL of K medium [9]. Microplates were incubated overnight at 20˚C with continuous shaking. The following day, arrested L1 animals were fed HB101 bacteria suspended in K medium. At the time of feeding, the

**Table 1. Suggested naming conventions for CellProfiler metadata.**

| Image_FileName_RawBF | Image_PathName_RawBF | Metadata_Date | Metadata_Experiment | Metadata_Plate | Metadata_Magnification | Metadata_Well |
|---|---|---|---|---|---|---|
| 20191119-growth-p05-m2X_C03.TIF | /CellProfiler/example/raw_images | 20191119 | growth | p05 | m2X | C03 |
| 20191119-growth-p06-m2X_C09.TIF | /CellProfiler/example/raw_images | 20191119 | growth | p06 | m2X | C09 |
| 20191119-growth-p09-m2X_C06.TIF | /CellProfiler/example/raw_images | 20191119 | growth | p09 | m2X | C06 |

The naming of "Metadata_Plate" and "Metadata_Well" are essential to the *setflags()*, *viewPlate()*, *viewWell()*, and *viewDose()* functions. Additionally, "Image_fileName_RawBF" and "Image_PathName_RawBF" are necessary for the proper function of *viewDose()*.

animals were also exposed to paraquat at one of six concentrations (0, 7.81, 31.25, 125, 500, 2000 μM) by serial dilution of a freshly thawed aliquot of 1.5 M paraquat solution. The final volume in each well after dosing and feeding was 75 μL. The animals were then grown for 48 hours at 20˚C with continuous shaking, afterwards the microplates were imaged to assess the effects of paraquat exposure on nematode development.

## Naming conventions

Several functions in the *easyXpress* package require specific naming conventions to work properly. For full details regarding essential file naming and directory structure see the package repository (https://github.com/AndersenLab/easyXpress). Importantly, when using the Metadata module in CellProfiler to extract information describing your images, specific column names are suggested (Table 1).

# Results

## Design and implementation

The easyXpress package is designed to be simple and accessible to users familiar with the R environment. The easyXpress package comprises nine functions for reading, processing, and visualizing large high-throughput image-based datasets acquired from microplate-based assays processed with CellProfiler (Fig 1). Because our software is built to handle CellProfiler data as

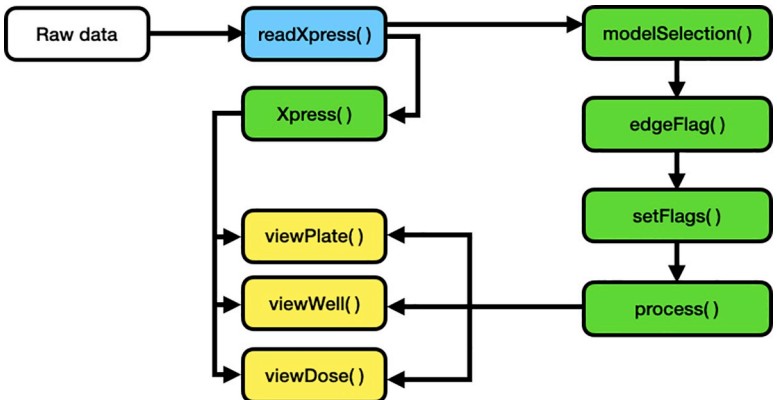

**Fig 1. easyXpress workflow.** The suggested workflow for using the easyXpress package starts with raw data generated from CellProfiler. For more information on implementing CellProfiler to generate data, see https://github.com/CellProfiler and https://github.com/AndersenLab/CellProfiler. Reading steps are shown in blue, processing steps are shown in green, and visualization steps are shown in yellow.

input, we suggest users review the overview and applications of CellProfiler as a prerequisite description of data generation [7]. Below, we describe the workflow for users to analyze their image data with easyXpress.

## Data import and model assignment

To read in CellProfiler data files, we provide *readXpress()*. Measurements calculated by Cell-Profiler can be exported in a comma-separated value (csv) file and accessed using *readXpress()*. For large-scale, high-throughput experiments, users can employ a computing cluster for increased analysis speed (https://github.com/AndersenLab/CellProfiler). In this case, CellProfiler data stored in.RData format is accessed using *readXpress()*. Additionally, the function can optionally import a design file created by the user containing experimental treatments and conditions. This design file is joined to the CellProfiler data and output as a single dataframe.

CellProfiler's WormToolbox detects and measures the phenotypes of individual animals based on user-calibrated models of variability in body size and shape [7]. To effectively detect animals in a mixed-stage population, multiple worm models must be used. However, using multiple worm models creates a one-to-many relationship between real animals and their measured phenotype (S1 Fig). We have included the function *modelSelection()* to annotate this information for downstream analysis. In instances where multiple worm model objects are assigned to a single primary object, *modelSelection()* will identify the best fitting model. Models are first ranked by frequency in the dataset such that the smallest model is classified as the most frequently occurring and the largest model is the least frequently occurring. In our experience, the most frequently occuring model in the dataset has the smallest size because it is often repeatedly assigned to a single primary object. Conversely, the least frequently occurring model in the dataset has the largest size as it is able to define the entire size of an animal, and is thus assigned to a primary object only once. The largest ranked model occuring within a single primary object is then selected as the best fitting model for that primary object. If necessary, *modelSelection()* will also specify whether the selected model object was repeatedly assigned to the same primary object and flag this event as a cluster. This problem occurs in instances where a model object is repeatedly assigned to a single primary object. If the largest model object is found to occur repeatedly in a single primary object, this model will be selected and a cluster flag will be added (S1C Fig). The *modelSelection()* step is essential to resolve cases where multiple instances of a selected model object are assigned to a single primary object, thus contributing to inaccurate phenotype measurements.

## Data pruning and summarization

Once the data are read into the R statistical environment, it is crucial to optimize data quality before in-depth analysis. Uneven well illumination can hinder the performance of CellProfiler's object identification and phenotype extraction. Despite correcting for uneven illumination within a well, discerning foreground objects from background can be especially challenging near the periphery of the well and can add noise to nematode phenotype data (S2 Fig). The function *edgeFlag()* was written to identify and flag animals located near the edge of circular wells using the centroid coordinates of the selected model object. By default, the function sets the radius of even illumination from the image center to 825 pixels, but this parameter can be adjusted by the user to serve project-specific analyses.

We also developed *setFlags()* in conjunction with *edgeFlag()* to further address data points that deviate from the expected animal measurements. The function *setFlags()* takes the output of *edgeFlags()* and detects outlier measurements among all measurements within a well using Tukey's fences [10]. By default, outlier calculations are performed by excluding data identified

by *modelSelection()* as part of a cluster as well as data in close proximity to the well edge. However, *setFlags()* is customizable, allowing the user to specify which filters to include. *edgeFlag()* and *setFlags()* were designed to allow for analysis-specific optimization when handling various experimental datasets. This flexibility in data filtering makes easyXpress extensible to many unique projects.

Once data are adequately flagged, the function *process()* organizes the data into a list containing four elements: raw data, processed data, and summaries for both datasets. The raw data element is the CellProfiler data following *modelSelection()* and flag annotation. The processed data are generated by default after subsequent removal of all cluster, edge, and outlier flags. If a user includes data annotated as clusters or edge cases in *setFlags()*, cluster and edge cases will be retained in the processed data output. Finally, it is often useful to summarize data by well to interpret patterns specific to experimental variables. Alternatively, measurements may be summarized by other experimental factors according to the individual experimenter's plate design. *process()* aids in the summarization of both the raw and processed data elements. This function comprehensively calculates the means, variances, quantiles, minimum, and maximum values of animal length for any experimental unit (*e.g.* well). We have also included the wrapper function *Xpress()* to accelerate the import and processing of CellProfiler data. *Xpress()* will perform the above functions with all default settings, but a user can alter input arguments to better suit project specific needs.

## Visualization

The easyXpress package provides several plotting functions to allow users to explore the data through detailed and elegant visualizations. After data summarization, it is often useful to inspect the values of the summary statistics in order to recognize patterns or identify potential outlier data. We provide *viewPlate()* to assist with the visualization of mean animal length within each well across a microtiter plate (Fig 2). This function accepts either raw or processed data to generate an interactive plot that allows users to scan across a plate to determine the number of objects present within individual wells.

To complement the top-level data visualization provided by *viewPlate()*, we have included *viewWell()* to allow users to deeply explore data within individual wells. This function

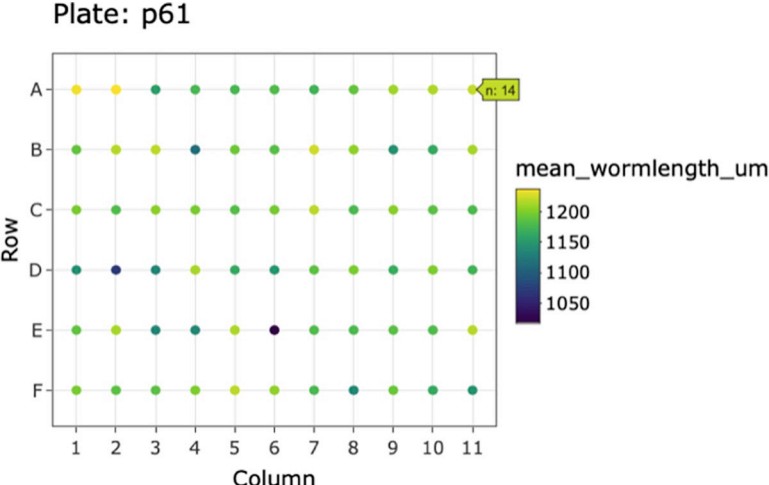

**Fig 2. Example plot generated by *viewPlate()*.** Well-wise plot of mean animal length (μm) from the summarized processed data. Interactive feature enables the assessment of the number of animals per well.

generates a plot of the well image following CellProfiler analysis with all objects annotated with their assigned class (Fig 3). Additionally, *viewWell()* can optionally generate a boxplot of the length values for each object. This plotting function is especially useful because it enables rapid qualitative assessment of object classification performance. By overlaying the model object classifications on the well image, users can quickly determine whether CellProfiler classified objects as expected or whether errors in model selection or data flags occurred.

Lastly, we have developed the function *viewDose()* to allow for the visualization of dose response data. *C. elegans* are often used to study conserved responses to various compounds [11–15]. *viewDose()* allows a user to visually examine the effect of a compound on animal size and shape over a range of concentrations (Fig 4). By specifying the strain and compound of

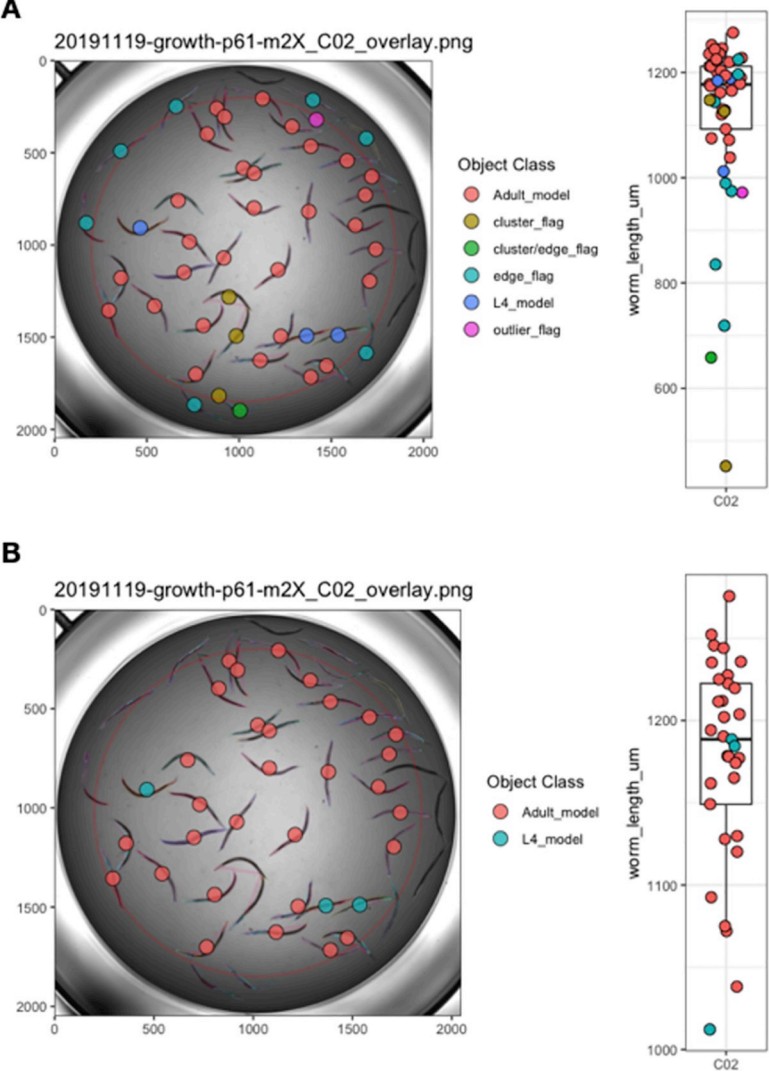

**Fig 3. Example plots generated by *viewWell()*.** The function *viewWell()* facilitates the exploration of data within an individual well. Well images displaying easyXpress raw (A) and processed (B) data are annotated with the location of each model object centroid (circles) and are colored by object class in the legend (left). Animals are outlined in different colors to indicate the model object(s) identified for each primary object (see S1 Fig). The length of each object is displayed as a boxplot (right). Well edge circumference defined by the function *edgeFlag()* is shown in red.

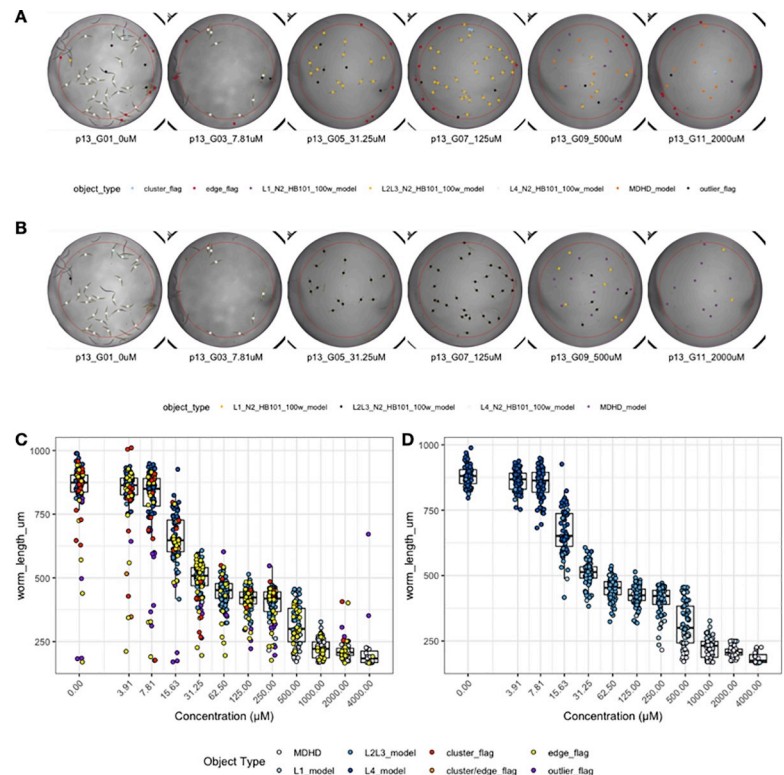

**Fig 4. Example plots generated by *viewDose()*.** The function *viewDose()* plots representative raw (A) or processed (B) well images with objects annotated by model class for each dose of a selected drug and strain. The length measurements of raw (C) and processed (D) are also shown.

interest, a plot of representative wells will be generated that includes labels for each identified object.

## Application to *C. elegans* growth data

We evaluated easyXpress using data collected from a *C. elegans* growth experiment [16]. Animals were imaged throughout the entire life cycle, beginning at the first larval (L1) stage and continuing until adulthood. Images were then processed with CellProfiler's WormToolbox and analyzed using easyXpress. During the implementation of easyXpress, four unique worm models representing *C. elegans* life stages were calibrated and applied: L1, L2/L3, L4, and Adult. These worm models do not designate stage assignments but rather represent the approximate sizes of animals that fall within the respective age groups (S1 Fig). The function *modelSelection()* assigned the appropriate model object to animals at each life stage, *edgeFlag()* and *setFlags()* identified outlier data points, and *viewWell()* provided clear visualizations of both the processed (Fig 5) and raw (S3 Fig) data.

## Conclusions

The easyXpress package presents an organized workflow for managing *C. elegans* phenotype data generated using CellProfiler. This package provides tools for the reading, processing, and visualization of these data in a simple and efficient way. By leveraging existing R infrastructure, easyXpress enables reproducible analysis, integration with other statistical R packages, and extensibility to many research projects using an open-source analysis pipeline.

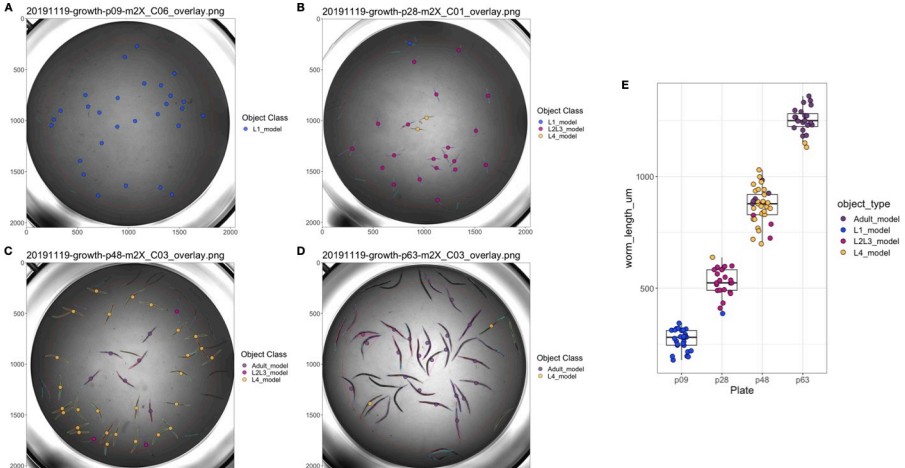

**Fig 5. easyXpress applied to *C. elegans* growth data.** A subset of well images acquired during *C. elegans* development displaying easyXpress processed data are shown here. Images taken at (A) 9 hours indicating the L1 stage, (B) 28 hours indicating the L2/L3 stage, (C) 46 hours indicating the L4 stage, and (D) 63 hours indicating the adult stage were analyzed with CellProfiler using four worm models. The easyXpress workflow was then used to process and visualize the data. The length of each object identified after processing is shown in (E).

## Supporting information

**S1 Fig. Multiple model objects assigned to a single primary object.** When running CellProfiler's WormToolbox with multiple worm models, multiple model objects can be assigned to a single primary object (real animal). Different colors are used to outline each worm model object. Here four unique models were used: L1, L2/L3, L4, and Adult. These worm models represent the approximate sizes of animals at each life stage. For example, some mutant or diverse wild genetic backgrounds might have differently sized adult animals as compared to the laboratory-adapted N2 strain. We have included this "soft matching" to account for small differences in the sizes of developmental stages across different genetic backgrounds, laboratories, and environmental conditions. (A) An animal detected by CellProfiler as a primary object has been assigned three unique worm models: two L1 model objects, one L2/L3 model object, and one L4 model object. *modelSelection()* classifies this animal as an L4 model object. (B) An animal detected as a primary object has been assigned four unique worm models: three L1 model objects, two L2/L3 model objects, one L4 model object, and one Adult model object. Here, *modelSelection()* identifies the Adult model as the best fitting model object. (C) An animal detected as a primary object has been assigned two unique worm models: three L1 model objects, and two L2/L3 model objects. In this case, *modelSelection()* classifies this animal as an L2/L3 model object and adds a cluster flag annotation to indicate the repeated assignment of the selected model object to the primary object.
(TIFF)

**S2 Fig. Uneven illumination along well edge hinders CellProfiler's ability to segment animals from background.** (A) Left is raw intensity values across well. (B) Right is with background correction. Intensities of object illumination are displayed on each z-axis. Objects near the edge of the well (y < 500 and y > 1500) have similar raw detected intensities (int) to more medial objects (y ~ 1000) in (A) but lower corrected intensities in (B) because of uneven background correction. Raw and background-corrected image segments are displayed in (C). Notice animals on the edges of the well do not stand out from the background as much as

animals in the center of the well and therefore are more challenging to discern.
(TIFF)

**S3 Fig. Raw data from *C. elegans* growth experiment displayed by the function *viewWell()*.**
Similar to Fig 5, well images taken at (A) 9 hours indicating the L1 stage, (B) 28 hours indicating the L2/L3 stage, (C) 46 hours indicating the L4 stage, and (D) 63 hours were analyzed. Here, the raw data results are displayed. The length of each identified object identified is shown in (E).
(TIFF)

# Acknowledgments

We would like to thank members of the Andersen laboratory for their helpful suggestions and feedback developing easyXpress.

# Author Contributions

**Conceptualization:** Joy Nyaanga, Timothy A. Crombie, Samuel J. Widmayer, Erik C. Andersen.

**Data curation:** Joy Nyaanga, Timothy A. Crombie, Samuel J. Widmayer.

**Formal analysis:** Joy Nyaanga, Timothy A. Crombie, Samuel J. Widmayer.

**Funding acquisition:** Erik C. Andersen.

**Investigation:** Joy Nyaanga, Timothy A. Crombie, Samuel J. Widmayer.

**Methodology:** Joy Nyaanga, Timothy A. Crombie, Samuel J. Widmayer.

**Project administration:** Erik C. Andersen.

**Resources:** Joy Nyaanga, Samuel J. Widmayer.

**Software:** Joy Nyaanga, Timothy A. Crombie, Samuel J. Widmayer.

**Supervision:** Erik C. Andersen.

**Visualization:** Joy Nyaanga, Timothy A. Crombie, Samuel J. Widmayer.

**Writing – original draft:** Joy Nyaanga.

**Writing – review & editing:** Joy Nyaanga, Timothy A. Crombie, Samuel J. Widmayer, Erik C. Andersen.

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
