## [Decision Letter · Decision Letter 0]

12 Jul 2021

PONE-D-21-15039

easyXpress: An R package to analyze and visualize high-throughput C. elegans microscopy data generated using CellProfiler

PLOS ONE

Dear Dr. Andersen,

Thank you for submitting your manuscript to PLOS ONE. THANK YOU FOR YOU PATIENCE! After careful consideration, we feel that it has merit but does not fully meet PLOS ONE’s publication criteria as it currently stands. Therefore, we invite you to submit a revised version of the manuscript that addresses the points raised during the review process.

Please submit your revised manuscript by  AUGUST 27, 2021. If you will need more time than this to complete your revisions, please reply to this message or contact the journal office at plosone@plos.org. Please include the following items when submitting your revised manuscript:

We look forward to receiving your revised manuscript.

Kind regards,

Heidi A. Tissenbaum

Academic Editor

PLOS ONE

Journal Requirements:

Reviewers' comments:

Reviewer's Responses to Questions

**Comments to the Author**

1. Is the manuscript technically sound, and do the data support the conclusions?

Reviewer #1: No

Reviewer #2: Yes

2. Has the statistical analysis been performed appropriately and rigorously? 

Reviewer #1: No

Reviewer #2: Yes

3. Have the authors made all data underlying the findings in their manuscript fully available?

Reviewer #1: No

Reviewer #2: Yes

4. Is the manuscript presented in an intelligible fashion and written in standard English?

Reviewer #1: Yes

Reviewer #2: Yes

5. Review Comments to the Author

Reviewer #1: Authors has developed a package easyXpress based on R-Statistical Environment platform that can facilitate the analysis with CellProfiler. This study has potential and could be beneficiary for many scientists looking for a solution. However, I find it difficult to assess the methods that they have developed due to limitation posed with unavailability of the code as well as statistical description of functions. In addition, I have few questions to authors as follows.

Q.1 : Authors mentioned that the code is available on github, however, I was unable to see the code or validate that it is working in the way it is proposed.

Q.2 Many place it is written to refer the manual however manual is not accessible, and information is not provided in the manuscript. My suggestion is, since it is package -based article, authors must provide the code to validate their claim or write full description in the manuscript.

Q.3 Authors mention that they have developed some functions to customize the filtering of noise in data, however no description or statistics underlying the function is explained anywhere in manuscript.

Q.4 I am unable to find the explanation/statistics for summarization in methods though it is mention.

Reviewer #2: This paper presents a piece of software meant to aid in the management and analysis of data first examined by CellProfiler. Beyond offering a method to visualize the outputs of CellProfiler, with the option of overlaying annotations, it provides further functionality in terms of refining annotations and removal of C. elegans deemed as outliers by custom filtering routines. Flexible control of functions and hyperparameters make this software amenable to an array of imaging situations. The contributions of this software, although limited when compared to a package such as CellProfiler itself, seem meaningful as it automates processes that are otherwise menial and time consuming. Further, inter-experimental comparisons will benefit from the consistency offered by using this software to manage how outlier removal occurs within a dataset. This paper makes meaningful contributions to the field of study however, there remains a few outstanding questions after reading the paper. Those questions can be found below.

Lines 130-143 begin to describe the modelSelection function, however, some items remain unclear. Based on the description starting at line 136, it seems to be that modelSelection would select the most infrequent model as the best model fit. Given that this is the correct interpretation, it seems to run against intuition where if one were to rely on model frequency alone to clear up cases of ambiguous labeling, the most frequent model may be better suited. If the least frequent is indeed the model that is chosen in cases of ambiguity, it would be helpful to elaborate on why this is the case. If this was understood incorrectly, it would be helpful to expand upon this sentence and provide further details on how models are ranked/selected.

The following sentence starting on line 139 raised another question. It seemed to be in the previous sentences that a "primary object" was referencing a C. elegans however, the statement "modelSelection() will also specify whether the selected model object was repeatedly assigned to the same primary object..." makes this conclusion seem unlikely. It is understood that a single C. elegans could be assigned multiple instances of different models however, it is unclear how a C. elegans could be assigned multiple instances of the exact same model. Due to this confusion, it would be helpful to have a statement on either i) what a primary object is in reference to, if it is not a C. elegans, or ii) in what cases one would expect a C. elegans to receive duplicates of the same model label.

The authors indicate that the easyXpress will help with necessary statistical tools for quality control and provide quantitative analysis of high-throughput imaging datasets. Although Figure 4 indicates a study using paraquat on worm measurements, the authors did not provide many details neither present quantitative results from this data. It is advisable to indicate the experimental details in the method section and show a detailed statistical analysis of this data. To show the full power of this platform, the author should represent the quantitative data of the study, worm body sizes, and the necessary statistical analysis.

Figure 3 identifies a few animals as L4_model while they appear to be of similar size as the adult animals as per the box plot on the right. Were these animal images confirmed with high-resolution imaging or manual inspection? If these annotations were erroneous, the authors are encouraged to discuss possible reasons for this error and whether it will affect the measurements.

6. PLOS authors have the option to publish the peer review history of their article (what does this mean?). If published, this will include your full peer review and any attached files.

Reviewer #1: No

Reviewer #2: No

---

## [Author Response · Author response to Decision Letter 0]

21 Jul 2021

Our responses are in the Responses document.

---

## [Decision Letter · Decision Letter 1]

2 Aug 2021

easyXpress: An R package to analyze and visualize high-throughput C. elegans microscopy data generated using CellProfiler

PONE-D-21-15039R1

Dear Dr.Eric Amdersom,

We’re pleased to inform you that your manuscript has been judged scientifically suitable for publication and will be formally accepted for publication once it meets all outstanding technical requirements.

As you can see the reviewer's were pleased with the revisions. Congratulations and thanks for your patience!

Kind regards,

Heidi A. Tissenbaum

Academic Editor

PLOS ONE

Additional Editor Comments (optional):

Reviewers' comments:

Reviewer's Responses to Questions

**Comments to the Author**

1. If the authors have adequately addressed your comments raised in a previous round of review and you feel that this manuscript is now acceptable for publication, you may indicate that here to bypass the “Comments to the Author” section, enter your conflict of interest statement in the “Confidential to Editor” section, and submit your "Accept" recommendation.

Reviewer #1: All comments have been addressed

Reviewer #2: All comments have been addressed

2. Is the manuscript technically sound, and do the data support the conclusions?

Reviewer #1: Yes

Reviewer #2: Yes

3. Has the statistical analysis been performed appropriately and rigorously? 

Reviewer #1: Yes

Reviewer #2: Yes

4. Have the authors made all data underlying the findings in their manuscript fully available?

Reviewer #1: Yes

Reviewer #2: Yes

5. Is the manuscript presented in an intelligible fashion and written in standard English?

Reviewer #1: Yes

Reviewer #2: Yes

6. Review Comments to the Author

Reviewer #1: Authors have answered all my questions and made available the code accessible. I have no further question.

Reviewer #2: All comments have been addressed. The paper is well written and provides a useful information to the community of C. elegans researchers running high-throughout screens.

7. PLOS authors have the option to publish the peer review history of their article (what does this mean?). If published, this will include your full peer review and any attached files.

Reviewer #1: No

Reviewer #2: **Yes: **Adela Ben-Yakar

---

## [Editor Report · Acceptance letter]

4 Aug 2021

PONE-D-21-15039R1 

easyXpress: An R package to analyze and visualize high-throughput *C. elegans* microscopy data generated using CellProfiler 

Dear Dr. Andersen:

I'm pleased to inform you that your manuscript has been deemed suitable for publication in PLOS ONE. Congratulations! Your manuscript is now with our production department. 

Kind regards, 

on behalf of

Dr. Heidi A. Tissenbaum 

Academic Editor

PLOS ONE